# North Atlantic warming over six decades drives decreases in krill abundance with no associated range shift

Martin Edwards [1✉], Pierre Hélaouët[2], Eric Goberville [3], Alistair Lindley[2], Geraint A. Tarling [4], Michael T. Burrows[5] & Angus Atkinson [1]

In the North Atlantic, euphausiids (krill) form a major link between primary production and predators including commercially exploited fish. This basin is warming very rapidly, with species expected to shift northwards following their thermal tolerances. Here we show, however, that there has been a 50% decline in surface krill abundance over the last 60 years that occurred in situ, with no associated range shift. While we relate these changes to the warming climate, our study is the first to document an in situ squeeze on living space within this system. The warmer isotherms are shifting measurably northwards but cooler isotherms have remained relatively static, stalled by the subpolar fronts in the NW Atlantic. Consequently the two temperatures defining the core of krill distribution (7–13 °C) were 8° of latitude apart 60 years ago but are presently only 4° apart. Over the 60 year period the core latitudinal distribution of euphausiids has remained relatively stable so a 'habitat squeeze', with loss of 4° of latitude in living space, could explain the decline in krill. This highlights that, as the temperature warms, not all species can track isotherms and shift northward at the same rate with both losers and winners emerging under the 'Atlantification' of the sub-Arctic.

[1] Plymouth Marine Laboratory, Plymouth PL13DH, UK. [2] The Marine Biological Association (MBA), The Laboratory, Plymouth PL12PB, UK. [3] Unité Biologie des Organismes et Ecosystèmes Aquatiques (BOREA), Muséum National d'Histoire Naturelle, Sorbonne Université, Université de Caen Normandie, Université des Antilles, CNRS, IRD, Paris, France. [4] British Antarctic Survey, Cambridge CB30ET, UK. [5] Scottish Association for Marine Science (SAMS), Scottish Marine Institute, Oban PA37 1QA, UK. ✉email: martin@pelasphere.com

Krill, lipid-rich zooplankton, are an important source of food for commercially exploited fish species, squid and other pelagic predators such as baleen whales and therefore represent a crucial element in North Atlantic food webs[1–3]. The boreal and Arctic waters of the North Atlantic are a major hot-spot for northern krill and at the basin scale it is estimated that the total stock biomass may be equivalent to that of Antarctic krill (*Euphausia superba*), presently estimated at 380 Mt[4]. Over the last few decades, krill (one of the few pelagic crustaceans to be commercially harvested) have become increasingly exploited as a source of protein for human and animal consumption leading to potentially detrimental effects for marine ecosystems and in particular to the ecology of their natural predators, including fish, seabirds and baleen whales[1,3,5].

Recently there have been two regional-scale, long-term studies looking at euphausiid trends centred around the Barents Sea and the Icelandic Basin[6,7]. Both studies concluded that hydroclimatic variability as well as biotic interactions were important factors dictating euphausiid abundance over decadal periods. Here we investigate multidecadal spatial and temporal patterns of krill over six decades at the ocean basin scale using data collected by the Continuous Plankton Recorder (CPR) Survey. We relate these spatial and temporal patterns to the warming climate of the North Atlantic and include natural climate variability such as the North Atlantic Oscillation (NAO).

## Results

Using all data collected by the CPR survey between January 1958 and December 2017 (see 'Methods'[8]), the total mean spatial distribution of euphausiids is shown in Fig. 1a. From the spatial analysis, the main centre of distribution of euphausiids in the North Atlantic is found in the colder waters of the sub-polar gyre to the south of Greenland and to the east of the Newfoundland shelf. Other significant populations are found along the continental European shelf edge, the Norwegian Sea and off the coast of Norway. While some populations are found in coastal and shallower regional seas such as the North Sea, the vast majority of euphausiids in the North Atlantic are found over deeper oceanic waters. It is thought that the krill abundance collected by the CPR survey mainly reflects the surface indices of the larval and juvenile stages of the three most common euphausiid species in the North Atlantic (see 'Methods') as well as the adult stages of the smallest species (*Thysanoessa longicaudata*) (see 'Potential biases and limitations').

Considering total euphausiids over time for the whole oceanic basin, Fig. 1b shows that abundances rise to a peak in July before decreasing by October. Over the whole 1958–2017 period, two clearly anomalous periods are seen, both associated with a decrease in abundance occurring in the late 1980s and from the mid-1990s. The mid-1990s decrease in abundance has continued to the present time-period, with low numbers throughout the last 20 years, in contrast to the high abundances from 1958 to the mid-1980s. It must be noted that the anomalous period found in the late 1980s occurs when the sampling effort (Fig. 1c) is lower than average. While lower sampling effort could in theory lead to an underestimation of euphausiid abundance, sampling effort during the second low period was not different from that during the high-abundance period. Further, specific measures were taken during the data processing to further mitigate the potential impact of sampling effort (see 'Methods').

Next, we examined the spatial distribution of euphausiids over six decadal periods (Fig. 2). Abundances were high for the first three time-periods, with a major stronghold in the NW Atlantic. Abundances remained high in this sub-polar gyre in the period between 1988 and 1997, but this decade marked a decline in the

North East Atlantic and, in particular, the North Sea. Note that this period had a lower sampling frequency in the central North Atlantic waters. However, sampling effort remained consistent for the sub-polar gyre region, apart from a brief hiatus in the mid-1980s, and the North East Atlantic reflecting real patterns in the time-series. During the last decade some of the CPR routes have extended to both a more southerly and northerly extent in the SW of the North Atlantic and into the Norwegian Sea. After the decline in the east, there was a subsequent major decline in abundance across the main population centre in the western area during 1998–2007 and 2008–2017. This abrupt decline in abundance since the mid-1990s can also be seen in Fig. 1b.

To further understand the spatial and temporal trends in the euphausiid populations of the North Atlantic we calculated the thermal preference/habitat (taxa temperature index) of euphausiids for each of the six decades: 1958–1967; 1968–1977; 1978–1987; 1988–1997; 1998–2007; 2008–2017 (Fig. 3a) represented by percentiles of the abundance. To evaluate whether the euphausiid populations are moving northward over a decadal period, we calculated the latitudinal range of euphausiids over the same decadal periods using percentiles of the abundance (Fig. 3b). Finally, we calculated the latitudinal position of key North Atlantic isotherms over the same time-periods (Fig. 3c).

According to taxonomic records (from CPR data), *Thysanoessa* spp. are becoming rarer in CPR samples over the last decade, particularly for the North East Atlantic and adjacent seas with *Thysanoessa raschii* becoming rare in the northern North Sea. The most common species recorded in the sub-polar region is *T. longicaudata* and the sub-polar region is where euphausiids have declined most in terms of numerical abundance according to the decadal biogeographical analysis and taxonomic records. The distribution of this dominant boreal species in the North Atlantic Ocean is primarily governed by the thermal regime with the upper limit of the species distribution ~15 °C and the lower limit between 2 and 3 °C[9] with the central core of the population (50% tile) centred around 8–9 °C (Fig. 3a, presumably representing the aforementioned species *T. longicaudata*). The core of the population in this study has a slightly lower temperature than those derived by a previous euphausiid study[7]. Examining the thermal envelopes per decade, the thermal preference of the core abundance of upper layer euphausiids have remained relatively stable through time particularly for the northern edge of the population core, centred around the 7 °C isotherm, however, the southern edge of the distribution population has been less stable (Fig. 3a). The lack of biogeographical decadal movement (Figs. 2 and 3b), however, suggests that the species has not carved out new northerly habitats (in the upper layers) but has declined in its geographical stronghold within the sub-polar gyre. The core latitudinal distribution of the species at ~55 °N has remained markedly stable over the 60-year period (Fig. 3b). This is in contrast to the many documented species northerly movements found in the North East Atlantic[10]. During the same period, some of the sea surface temperature (SST) isotherms in the North Atlantic have shown considerable movement particularly the more southerly warmer surface isotherms 12–13 °C (Fig. 3c). The differences in the pace of northerly movement of the warmer isotherms compared to the colder isotherms has changed the original difference of 8° of latitude between the 12–13 °C isotherm and the 7–8 °C isotherm for the first decadal period (1958–1967) to 4° of latitude for the most recent time-period (2007–2017, Fig. 3c.). This has led to a contraction of available core habitat of ~4° of latitude over the 60-year period.

To examine the dominant trends in the euphausiid data in both space and time, and its relationship with the climate of the North Atlantic, we used a standardised principal component analysis (PCA) to identify the main spatial patterns and its associated

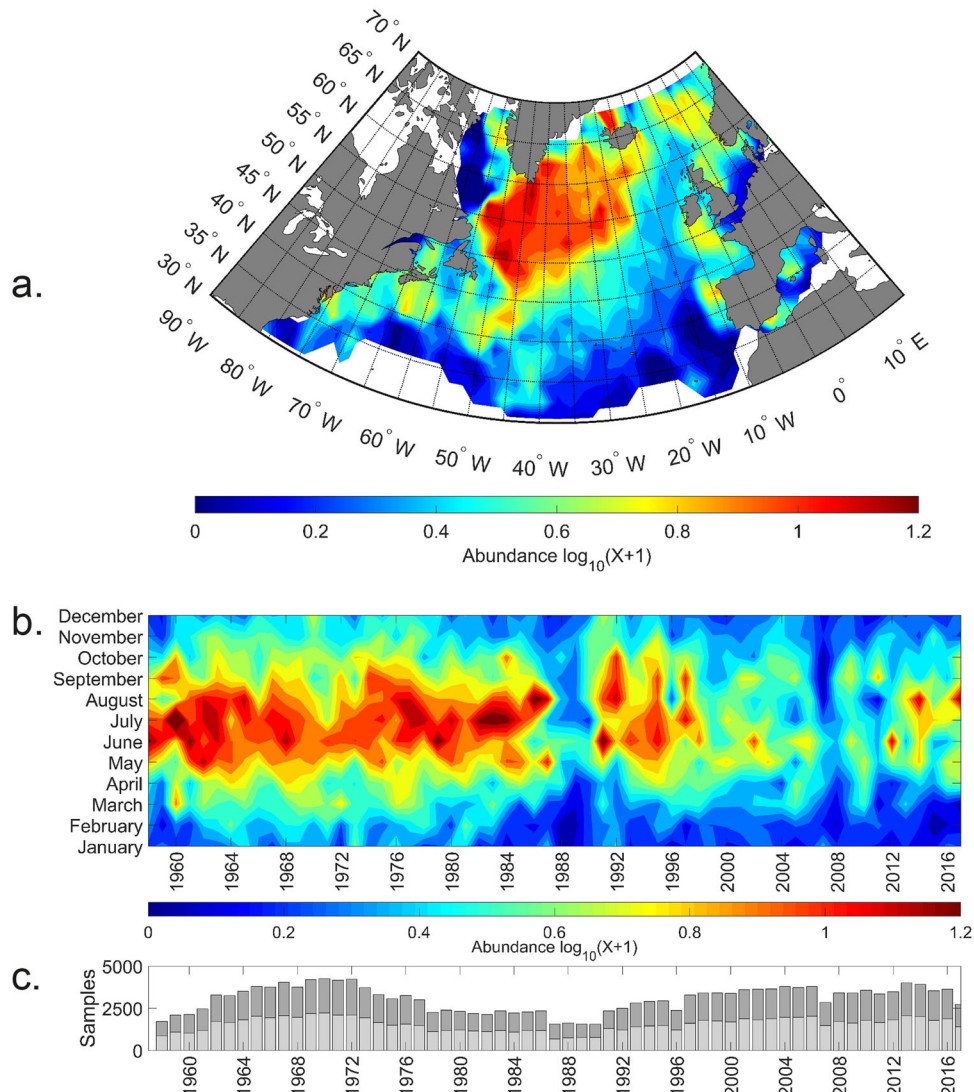

**Fig. 1 Geographical and long-term patterns in euphausiids in the North Atlantic. a** Mean spatial interpolated distribution of euphausiids in the North Atlantic. **b** Mean monthly abundance of euphausiids in the North Atlantic from 1958–2017 based on interpolated data. **c** Number of samples used to calculate euphausiid abundance in the North Atlantic; light shading: day samples, dark shading: night samples.

dynamic through time. When mapped, the resulting eigenvectors describe the spatial pattern of correlations between changes in euphausiid abundance and the main temporal pattern described by the first principal component (PC1). To examine a potential relationship with the North Atlantic climate, this analysis was conducted on both euphausiids and SST data. The resulting eigenvector maps and the time-series associated with the eigenvector (known here as the principal component time-series (PC-TS) are shown in Fig. 4). The first PC-TS for total euphausiids (18.56% of the total variance) for the North Atlantic shows a rapid and abrupt change beginning in the late 1980s and particularly prevalent after the mid-1990s. The first PC-TS for SST (17.51% of the total variance) and its resulting eigenvector map for the North Atlantic is shown in Fig. 4c, d. The main trend in SST for the North Atlantic (excluding the first four years) is a general decline in temperature followed by an abrupt increase in temperature seen since the mid-1990s. This is the general pattern observed in the North Atlantic Ocean and is associated with climate warming and trends in Atlantic multidecadal variability[11]. At the whole North Atlantic basin scale, inter-annual correlations

between the first PC-TS of euphausiid abundance and (i) the first PC-TS of SST, (ii) the Atlantic Multidecadal Oscillation (AMO) index and (iii) the NAO index were $r = -0.702$ ($p_{ACF} < 0.01$), $r = -0.63$ ($p_{ACF} < 0.1$) and $r = -0.392$ ($p_{ACF} < 0.1$), respectively. Highest multiple correlations at the whole North Atlantic and 60-year time scale were found between euphausiid abundance and SST plus an estimate of total phytoplankton biomass (over the same time-period) from the CPR survey ($r = 0.826$, $p_{ACF} < 0.1$). The general macro trend in phytoplankton biomass since the 1960s is an increase in biomass in the sub-polar North Atlantic and a decrease in phytoplankton biomass in the sub-tropical gyre regions. This is consistent with the hypothesis of a decrease in phytoplankton in sub-tropical regions due to enhanced stratification (nutrient limitation) and an increase in sub-polar regions due to a decrease in light limitation[12].

## Discussion

In the North Atlantic, at the ocean basin scale, there has been a significant evolution in climate due to natural variability and climate warming over the duration of this 60-year study. Many

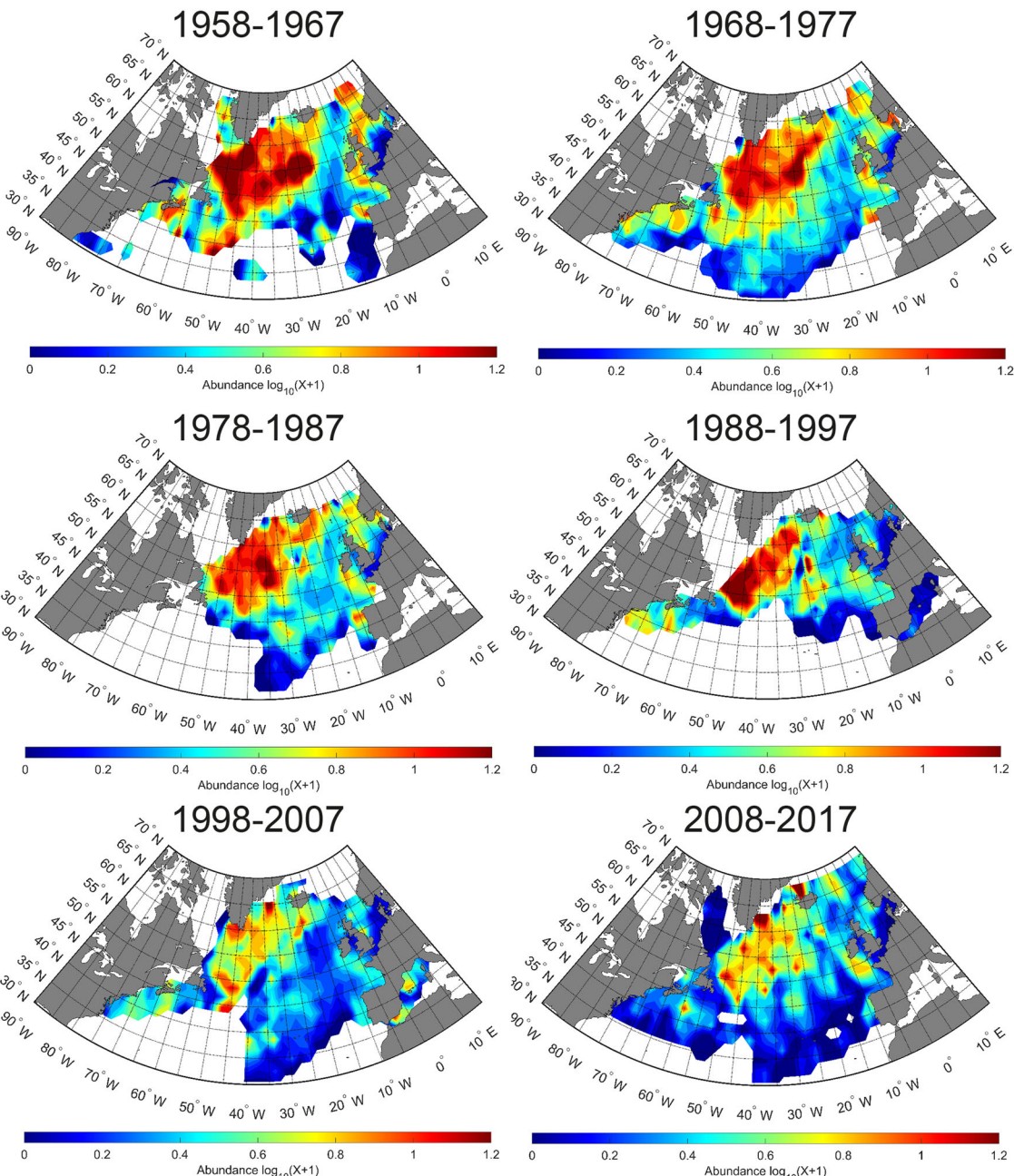

**Fig. 2 Mean spatial abundance of euphausiids per decade in the North Atlantic.** Mean spatial abundance of euphausiids per decade in the North Atlantic from 1958–2017 split into six time-periods 1958–1967; 1968–1977; 1978–1987; 1988–1997; 1998–2007; 2008–2017.

changes in marine biological populations and communities have been associated with Northern Hemisphere Temperature (NHT) trends, Atlantic multidecadal variability (AMV), the East Atlantic Pattern (EAP) and variations in the NAO index[11]. It has been estimated that ~50% of these biological changes are related to natural decadal climate variability and the other due to forced anthropogenic warming[13]. These have included changes in species distributions and abundance, the occurrence of sub-tropical species in temperate waters, changes in overall plankton biomass and seasonal length, and changes in the ecosystem functioning and productivity of the North Atlantic[14–16]. Over the last six decades, there has been a progressive increase in the presence of warm-water/sub-tropical species into the more temperate areas of the North East Atlantic and a decline of colder water species. The

mass biogeographical and rapid movements (~23 km.yr$^{-1}$) have been mainly related to movement in SST isotherms[10].

Overall, the total abundance of euphausiids in the North Atlantic has declined by more than 50% over the 60-year study period. Moreover, this decline has not been linear and two prominent abrupt shifts are clearly visible in the time-series (Fig. 1); the first occurring in the mid to late 1980s and another occurring after 1996. This large decline will presumably have had major consequences for the rest of the marine food web and will have important implications for ongoing fisheries management. Examining the different decadal spatial maps (Fig. 2), the main abrupt decline in the mid to late 1980s is predominantly seen in the North East Atlantic (Fig. 2, decadal map 1978–1987) and the shift that occurred after 1996 is found in the North West Atlantic

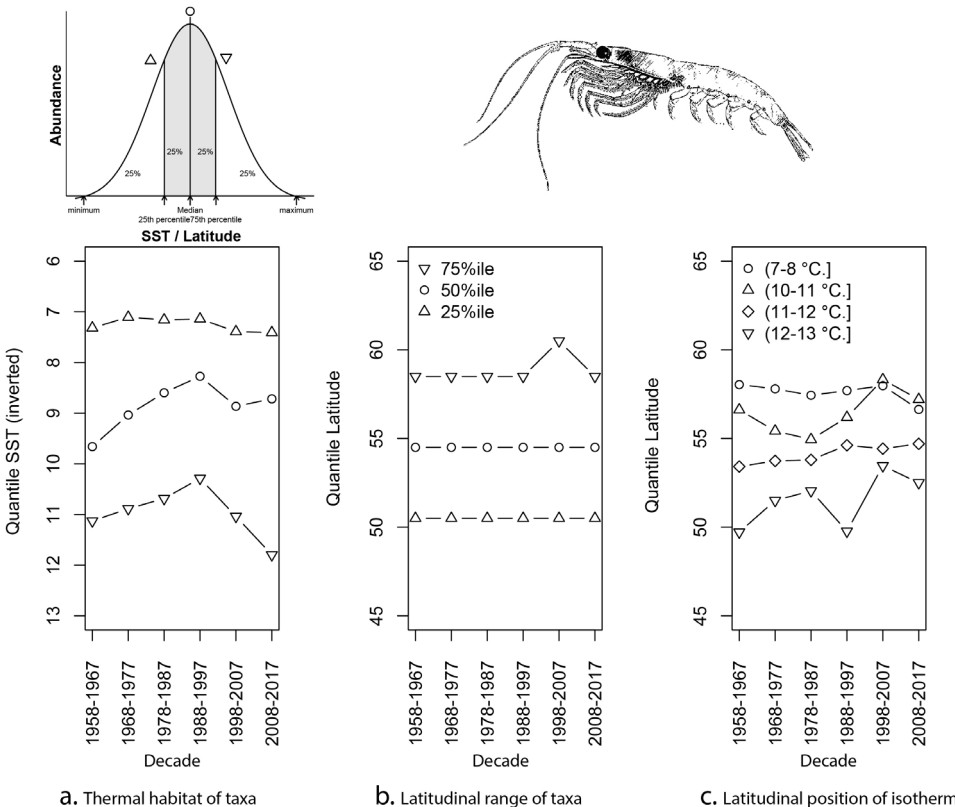

**Fig. 3 Multidecadal changes in the thermal habitat and latitudinal distribution of euphausiids in the North Atlantic from 1958–2017. a** Relationship between the gridded abundance and sea surface temperature (SST inverted) for the North Atlantic presented as weighted quantiles of the population for the decadal periods 1958–1967; 1968–1977; 1978–1987; 1988–1997; 1998–2007; 2008–2017. **b** Latitudinal range of euphausiids in the North Atlantic presented as weighted quantiles for the six decadal periods. **c** Latitudinal position of the SST isotherms 7–8 °C, 10–11 °C, 11–12 °C and 12–13 °C over the six decadal periods. Data from CPR sampling between 30–68 °N and 90 W–15 °E.

and, in particular, the sub-polar gyre region (Fig. 2, decadal map 1998–2007). Looking more closely at these two regions (Fig. 5), we can see that the main abrupt shifts occurred in the North Sea region after the late 1980s (total of 63% decline over the whole time-series) and the sub-polar time-series after 1996 (total of 43% decline over the whole time-series). These geographical differences in timing can be explained partly by the spatially heterogeneous responses of the biology to the different climate indices and climate variability of the North Atlantic. The North Sea abrupt change is associated with the well-documented ecological regime shift that occurred here at this time[17] related with trends in the NAO index and the geographical movement of the 10° thermal boundary[14]. The main abrupt shift after 1996 found in the other regions of the North Atlantic is associated with the main shift to a warm-phase in Atlantic conditions that occurred after 1996[11].

While there has been a large decrease in euphausiids over the last 60 years in the sub-polar region, other taxa in this region have remained relatively stable in time with some showing an increase. Data from the CPR survey show the dominant large copepod *Calanus finmarchicus* has shown a small decline, whereas the other dominant large copepods, *Metridia lucens*, *Medridia longa* and *Paraeuchaeta norvegica*, have shown increases since the 1960s. The pelagic hyperiids (amphipoda), forming a large proportion of the zooplankton biomass and third only to copepods and euphausiids in terms of biomass in the sub-polar gyre, have shown an opposite trend to the euphausiids with a 15% increase since the 1960s. Another important group of zooplankton, the appendicularians, have shown a dramatic increase, nearly quadrupling their abundance since the 1960s, suggesting that, while there has been an overall increase in phytoplankton biomass in this region, there could also be a trend towards a smaller size-fraction of phytoplankton. It is unclear why the euphausiids alone among the most dominant zooplankton taxa in this region have shown a particular decline since the 1990s. In contrast, in the North Sea, it has been widely documented that most boreal and cold-temperate species have declined over the last 60 years, particularly since the late 1980s, and have been replaced by more warm-water and temperate species[14,15,18]. For example, the boreal copepod *C. finmarchicus* has decreased by 50% in the North Sea since the late 1980s regime shift. High abundances of *C. finmarchicus* and euphausiids in the 1960s and 1970s have been associated with the North Sea gadoid outburst and their subsequent decline since the late 1980s have been associated with poor cod recruitment[19].

Euphausiid species inhabit productive regions worldwide, and this group appears particularly susceptible to climatic forcing and change. Similar to euphausiids in the sub-polar gyre, Antarctic krill, *E. superba*, has also experienced a decline in abundance in recent decades within the rapidly warming Atlantic sector[20], accompanied by a contraction in its distributional range towards the Antarctic continent[21]. The causes of this decline still require study, but potential factors include a decrease in sea-ice in important nursery regions and increasing temperatures towards its northerly limits[22]. Euphausiids are also important in the well-studied California Current region[23] where, they showed a rhythmic but complex relationship with the El-Nino/La Nina climatic cycle, with different species exhibiting different patterns

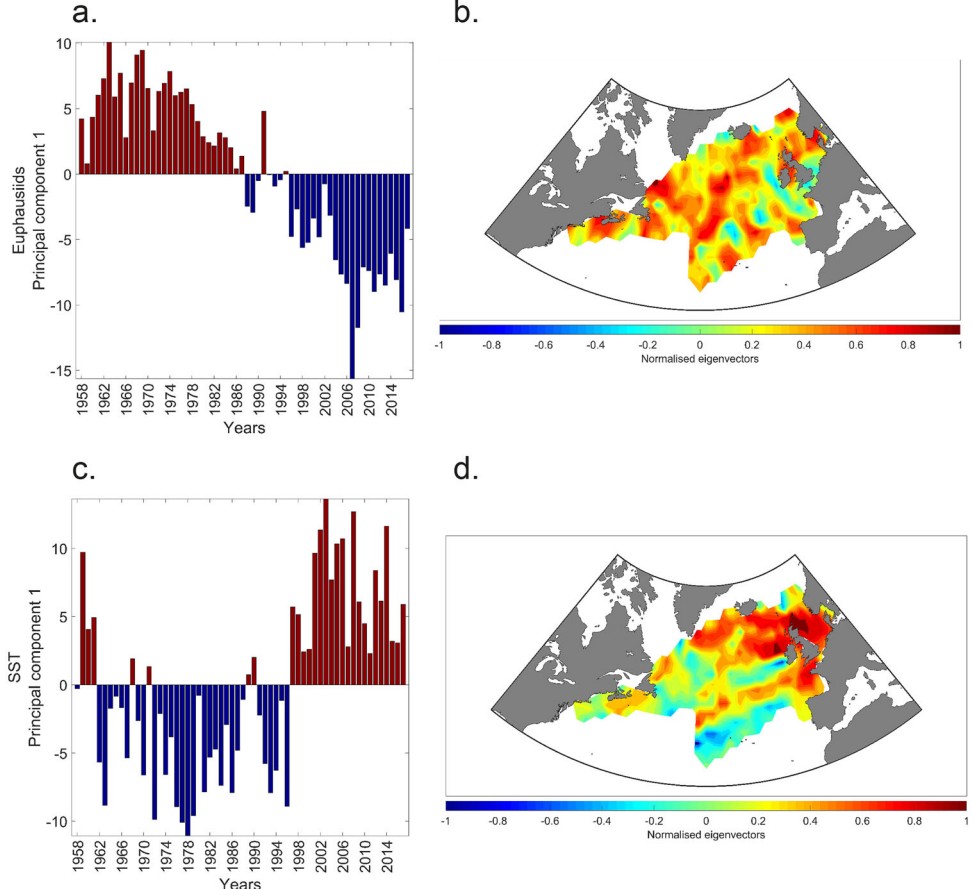

**Fig. 4 Long-term (1958–2017) changes in euphausiids and sea surface temperature in the North Atlantic. a** First principal component time-series (PC-TS) of euphausiids for the North Atlantic. **b** First normalised eigenvectors showing the spatial pattern of correlations between changes in abundance and the first PC-TS of euphausiids. **c** First PC-TS of SST for the North Atlantic. **d** First normalised eigenvectors showing the spatial pattern of correlations between changes in SST and the first PC-TS of SST.

of periodicity[24]. Nevertheless, over 50 years of study, there has been no evidence of any large-scale decline in any species.

It must be noted that due to the limitations of the sampling (restricted to 10 m depth) we cannot absolutely refute the hypothesis that euphausiid populations have simply adopted a deeper distribution to avoid the higher temperatures. However, we think this is unlikely due to that residing in the non-productive layers would make it extremely challenging for euphausiids to meet their energy budget. Hence, euphausiids avoiding the upper mixed layer and adopting a deeper distribution would be at a severe disadvantage and would likely become locally extinct over time. Also, although the thermal regime may have changed over recent decades, the day/night cycle, which acts as both a cue and a driver for their various diel vertical migration (DVM) patterns, has not changed and we expect this behaviour to remain the same. This means that any sampling bias caused by DVM will be the same across decades irrespective of any thermal shift.

In the sub-polar regions of the North Atlantic, where euphausiids are most abundant, concern is growing at the accelerated pace of these changes and the increasing 'Atlantification' (i.e. warmer more saline Atlantic waters) of these more northern waters and their detrimental effects on Arctic systems[25]. These have been undergoing a rapid transformation over the last few decades which includes loss of sea-ice cover and ocean warming. The Arctic sea regions, in particular, are experiencing the strongest warming on the planet (twice as fast as the planetary

average) and the loss of sea-ice in recent decades has been rapid[26]. Many regional seas that were once considered as being inhabited exclusively by Arctic flora and fauna have become more influenced by more southerly species as these species move northward as the Arctic warms. Recently, we have also witnessed the first transarctic migration of a plankton species in modern times[27].

As ocean temperature rises, we generally expect species distributions to track towards historically cooler regions in line with their thermal affinities[28]. However, this study shows for the first time in the North Atlantic that marine populations may not simply just shift their distributions northward due to shifting isotherms and hence re-establish new geographic habitats (thereby maintaining their relative abundance) but may in fact be spatially constrained due to ocean currents and strong thermal boundaries such as the polar front limiting their northward expansions[29]. As a consequence, a species may decline in situ inside its historical core geographical ranges and thermal niche as its habitat becomes spatially squeezed. In the case of euphausiids in this study, the lack of biogeographical decadal movement suggests that the species has not carved out new northerly habitats but has declined in its geographical stronghold in the sub-polar gyre. The core latitudinal distribution of the species at ~55 °N has remained markedly stable over the 60-year period. We have also shown that the isotherms for the warmer temperatures are shifting steadily northwards, while the cooler isotherms remain in place (Fig. 3) with an 8° difference in average latitudes of the 7–8 °C and 12–13 °C

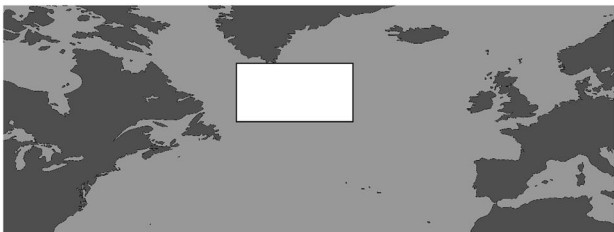

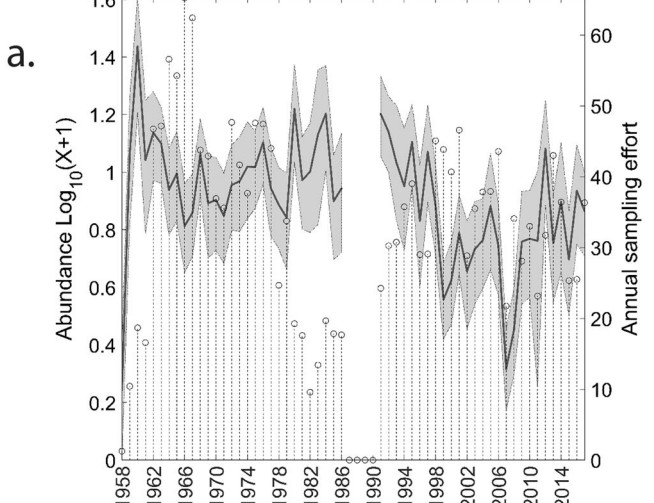

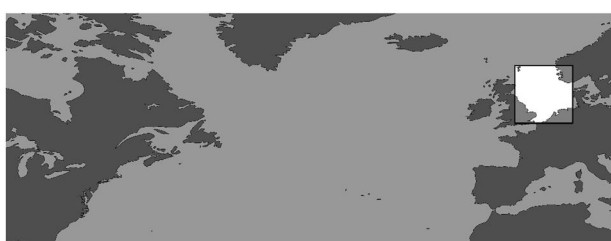

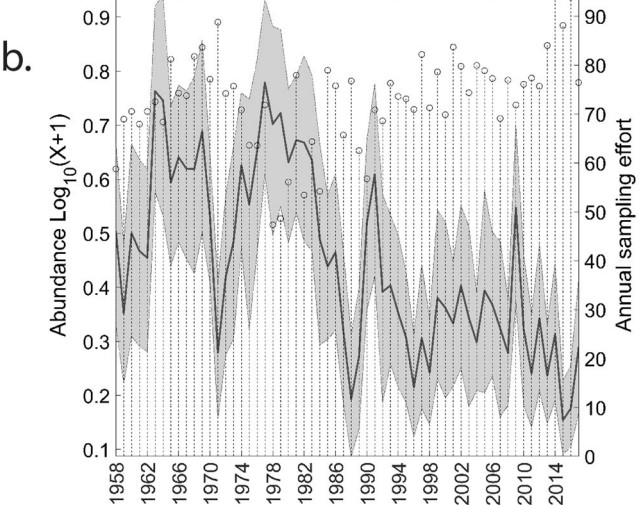

**Fig. 5 Long-term (1958–2017) changes in euphausiid abundance.**
Percentiles of bootstrapped mean abundances are represented: 5% lower line, 95% upper line, 50% bold line. **a** In the sub-polar gyre region of the North Atlantic from 1958–2017. **b** In the North Sea from 1958–2017. Grey shaded area. For both areas the sampling effort is overlaid (vertical dashed lines).

isotherms in 1958–1967 but only 4° of latitude between the same temperatures in 2008–2017. This 'habitat squeeze' and a potential habitat loss of 4° of latitude could be the main driver in the decline of euphausiid populations seen in this study. This study illustrates the potential for losers as well as winners under warming, which is partially dependent on the location of the population centres. However, temperature alone does not necessarily explain all patterns observed in this study and trophic interactions would also play an important role. We must also emphasise that the mechanisms for some the patterns observed cannot be fully explained by temperature and would warrant further investigation particularly at the regional scale.

The northward movement of isotherms and associated movement of thermally bound habitats will, therefore, not be a uniform process with isotherms moving slowly in regions with strong frontal boundaries and rapidly across regions with wide SST gradients. For example, while the North East Atlantic has potentially unimpeded northward flow of North Atlantic water into the Norwegian and Barents Seas having low SST gradients (seen as recent 'Atlantification' of these regions) leading to potential habitat expansions, the North West Atlantic is latitudinally stalled by the sub-polar gyre circulation which is geographically and temporally more robust and forms a thermal barrier to rapid expansion. This may help to explain the major, 50% decline in euphausiid populations across the North Atlantic, with major implications for the rest of the food web.

## Methods

**Biological data**. The CPR survey is a long-term, sub-surface marine plankton monitoring programme consisting of a network of CPR transects towed monthly across the major geographical regions of the North Atlantic. It has been operating in the North Sea since 1931 with some standard routes existing with a virtually unbroken monthly coverage back to 1946[30,31]. The CPR survey is recognised as the longest sustained and geographically most extensive marine biological survey in the world[15]. The dataset comprises a uniquely large record of marine biodiversity covering ~1000 taxa over multidecadal periods. The survey determines the abundance and distribution of phytoplankton and zooplankton (including fish larvae) in our oceans and shelf seas. Using ships of opportunity from ~30 different shipping companies, it obtains samples at monthly intervals on ~50 transocean routes. In this way, the survey autonomously collects biological and physical data from ships covering ~20,000 km of the ocean per month, ranging from the Arctic to the Southern Ocean.

The CPR is a high-speed plankton recorder that is towed behind 'ships of opportunity' through the surface layer of the ocean (~10 m depth)[8]. Water passes through the recorder and plankton are filtered by a slow-moving silk (mesh size 270 μm). A second layer of silk covers the first and both are reeled into a tank containing 4% formaldehyde. Upon returning to the laboratory, the silk is unwound and cut into sections corresponding to 10 nautical miles and approximately 3 m³ of filtered sea water[32]. There are four separate stages of analysis carried out on each CPR sample, with each focusing on a different aspect of the plankton: (1) overall chlorophyll (the phytoplankton colour index; PCI); (2) larger phytoplankton cells (phytoplankton); (3) smaller zooplankton (zooplankton 'traverse'); and (4) larger zooplankton (zooplankton 'eyecount'). The collection and analysis of CPR samples have been carried out using a consistent methodological approach, coupled with strict protocols and Quality Assurance procedures since 1958, making the CPR survey the longest continuous dataset of its kind in the world. Zooplankton analysis is carried out in two stages using CPR samples, with small (<2 mm) zooplankton identified and counted on-silk (representing ~1/50 of the filtering silk) and larger (>2 mm) zooplankton enumerated off-silk which includes a count of total Euphausiids that is used in this current study. The total euphausiid count recorded by the CPR includes adult stages as well as furcilia. The smaller stages such as nauplius and calyptopis are counted in the traverse analysis but are not included in this analysis.

**Euphausiid species**. In the North Atlantic the diversity of euphausiids is quite low with certain regional areas dominated by only a few species[33,34]. Taking the northern North Atlantic Ocean as a whole, the most numerically abundant near-surface species is *T. longicaudata*[35] with the species limited mainly to the boreal waters of the North Atlantic. The geographical distribution of *T. longicaudata* and aspects of the population dynamics of the species have been previously described[36]. In other regions of the North Atlantic, such as the Norwegian Sea, the species *Meganyctiphanes norvegica* (commonly known as northern krill) can dominate.

Although euphausiids are not speciated in routine analysis, previous studies using CPR data have identified the main species caught by the CPR. From those CPR studies, the dominating species in the boreal waters of the North Atlantic was *T. longicaudata*. This species was found to be present in both day and night samples with no significant difference between the two time-periods. The diurnal migration of this species has been found to be much less marked than in other species[36]. Other species that have been speciated from CPR samples include *Thysanoessa inermis* and *T. raschii* that mainly occur over the continental shelf with *T. raschii* particularly favouring more shallower waters, including the North Sea[37]. From the CPR survey, *M. norvegica* (northern krill) is mainly found over continental slopes, oceanic ridges and deep continental shelf waters including fjords[38]. Further south in the North Atlantic, the CPR regularly records the species *Euphausia krohnii*, *Nematoscelis megalops* and *Thysanoessa gregaria*. These species are characteristic of temperate waters and the transition zone between subarctic and more boreal and temperate waters of the North Atlantic[38].

**Night and day sampling**. Most epipelagic and a few mesopelagic krill species undergo diel vertical migrations (DVM), for example, both the species *T. longicaudata* and *M. norvegica* are typically found at depths of 100–400 m during the day and only found in the surface layers (0–100 m) during the night[9]. Figure 1c also shows the mean number of samples used in the analysis and divided between night and day samples. For this reason, we also treated and mapped day and night samples separately (Supplementary Fig. 1). Throughout the time-series, the number of night and day samples remains approximately equal and constant through time. To determine whether samples were taken by night or by day, spatio-temporal information of each sample were used to calculate their solar elevation (ranging from −80.2° to 81.2°). To maximise the difference between night and day biological communities, we have decided to remove all samples having a solar elevation ranging between −10° and 10°; 10° below the horizon (i.e. −10°) being an intermediate value between the ones used to characterise the nautical and civil twilight (−12° and −6°, respectively). Samples were grouped as follows: all samples = 258,919 or 100%, night samples = 97,338 or 37.6%, day samples = 100,786 or 38.9%, unused samples = 23.5%). The information on night/day krill species distribution was then compiled from the retained georeferenced (i.e. longitude and latitude) and time-referenced (year, month and day) observations.

**Regularisation of biological data**. To map spatial abundance (from 1958 to 2017, and per decade) and to estimate long-term changes in euphausiids in the North Atlantic, biological data were regularised on a 2° × 2° grid at a monthly scale for the period 1958–2017 using an inverse distance weighting (IDW) interpolation. A summary of the monthly CPR transects and their frequency can be found in[32]. A maximum search radius of 250 km was set, as well as a minimum number of neighbours of 5. The obtained gridded products have a total geographic size of 20 longitudes × 60 latitudes.

**Regularisation of SST data**. The Sea Surface Data originated from the Hadley Center[39]. Temperatures are reconstructed using a two-stage reduced-space optimal interpolation procedure, followed by superposition of quality-improved gridded observations onto the reconstructions to restore local detail. The selected dataset (HadISST1) is a 1° × 1° product that had to be re-gridded to match our regularised biological dataset, following a grid of 2° longitude × 2° latitude (i.e. 20 longitudes × 60 latitudes) for each year of the period 1958–2017, using a 2D interpolation using the nearest sample grid point.

**Statistics and reproducibility**. Standardised PCAs were performed at the annual scale on the deployed 3-way matrix, 60 years (period 1958 to 2017) × (20 longitudes × 60 latitudes) to investigate the major long-term changes in both the euphausiids community and SST. By segregating the variance into linearly orthogonal variables, PCAs allowed (1) the characterisation of the major temporal changes in both euphausiid abundance and SST and (2) the recognition of regions mainly influenced by the temporal patterns (i.e. mapping of the associated normalised eigenvectors).

*Statistical analysis of thermal habitat and isotherms*. To compare the changing distribution of krill with the changing distribution of temperature, temperature and latitude percentiles of krill abundance and 1 °C temperature classes were calculated per decade from 1958 onwards. The analysis was restricted to the geographical cells (i.e. 248 cells) that were sampled in all six decades to avoid inferred shifts being influenced by changes in sampling across latitudes among decades. Lower quartile (Q1, 25% of all observations), median (50% of all observations) and upper quartile (Q3, 75% of all observations) of temperature per decade for krill abundance were produced using the weighted quantile function in R (wtd.quantile in Hmisc library), with cell-specific krill abundance as weights. These values revealed changes in krill thermal habitat (Fig. 3a): advances or retreats of krill populations towards the warm-end of its distribution (changes in Q3); shifts relative to temperature for the centre of the abundance distribution (changes in the median); shifts relative to temperature in the cool part of the range (changes in Q1).; no change implies that krill distributions tracked changes in isotherms across decades. Quartiles (Q1, median and Q3) of latitude for krill abundance were similarly produced, with

cell-specific krill abundance as weights. Changes in krill-abundance-weighted latitude quartiles showed whether krill has shifted poleward in the cold-half, middle or warm-half of their range, while no change in the metrics is equivalent to an absence of geographical range shift. Median latitudes of SST classes for the same 248 cells sampled in every decade were produced for SST in separate 1 °C categories. Shifts in median latitudes of SST classes showed how the distribution of temperatures has changed across decades. Increasing median latitudes would be associated with poleward shifts in the corresponding isotherms.

*Potential biases and limitations*. The CPR survey currently records ~1000 plankton entities (many to species level) in routine taxonomic analysis with strict Quality Assurance protocols in operation for its sampling and plankton counting procedures since 1958. Due to the mesh size of CPR silks, many plankton species are only semi-quantitatively sampled owing to the small size of the organisms. In the case of phytoplankton there is thus a bias towards recording larger armoured flagellates and chain-forming diatoms and that smaller species abundance estimates from cell counts will probably be underestimated in relation to other water sampling methods. However, the proportion of the population that is retained by the CPR silk reflects the major changes in abundance, distribution and specific composition, i.e. the percentage retention is roughly constant within each species even with very small-celled species[40]. Similarly, potential underestimation of zooplankton abundances has recently been thoroughly statistically explored which found that while the CPR survey does underestimate abundance in some cases the CPR survey does give a correct picture of both temporal (i.e. seasonal and diel scales) and spatial (i.e. regional- to basin-scale) changes in zooplankton taxa in this case the species *C. finmarchicus*[41]. The study also showed that while the CPR sampling is restricted to the surface waters ~10 m in depth the seasonal and diel patterns of abundance of *C. finmarchicus* were positively correlated to patterns of abundance to a depth of 100 m[41].

While all sampling and analytical tools for measuring plankton have their own individual strengths and weaknesses, the CPR survey has had perhaps the most well-documented and transparent examination of its consistency and compatibility of any ecological time-series over time[15,30]. Perhaps the most obvious limitation of CPR sampling is its underestimation of components of the plankton, e.g. large-plankton-like fish larvae and delicate gelatinous plankton. This has been well-documented and users of the data are advised of the CPR's potential biases[32]. It is also widely recognised that all plankton sampling systems have their own limitations and nuances and all underestimate abundance to some degree and that the varying mechanisms are not always directly comparable[42]. However, in the case of krill, as in this study, the CPR may be underestimating the larger species due to the small sampling aperture (12.7 × 12.7 mm) of the CPR mechanism. As estimated in Silva et al., 2014[7], ~87% of the larvae and juveniles of krill are captured by the CPR but it is less efficient at catching larger krill, therefore, the CPR krill data may not represent absolute abundances of adult krill but will provide indices of the larval and juvenile stages of the three most common euphausiid species in the North Atlantic as well as the adult stages of the smallest species (*T. longicaudata*). It is likely that the large adult stages of the euphausiid species *M. norvegica* are underestimated in CPR sampling. Taking the northern North Atlantic Ocean as a whole, as well as within the specific area of the present study, the most numerically abundant near-surface species is *T. longicaudata*, making it likely that our findings are representative of real changes in euphausiid populations. However, further north, in areas such as the Barents Sea (not covered in this study), where the species *M. norvegica* can dominate, the CPR sampling is less likely to reflect real changes in the adult euphausiid populations. It is not known whether certain taxa such as krill are capable of gear avoidance and in some cases this might be a bias for some fast-moving taxa, however, since CPR sampling is conducted at high speeds (up to 20 knots), gear avoidance is considered to be fairly minimal[42]. A detailed study has been conducted on flow rate and ship speed on CPR sampling[43] given that the speed of the ships has, in some circumstances, increased since the 1960s, which may impact sample efficiencies. However, no significant correlation was found between the long-term changes in the speed of the ships and two commonly used indicators of plankton variability: the Phytoplankton Colour and the Total Copepods indices. This absence of relationship may indicate that the effect found is small in comparison with the influence of hydroclimatic forcing[43]. For further details on the technical background, methods, consistency, and comparability of CPR sampling, see[30].

**Reporting summary**. Further information on research design is available in the Nature Research Reporting Summary linked to this article.

## Data availability

The biological data that support the findings in this study are available from the open data portal. See https://www.mba.ac.uk/blog/oceans-open-data for more information. The Sea Surface Data originated from the Hadley Center. See HadISST1.

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

## Acknowledgements
An international consortium of government agencies from Canada, Norway, UK and USA funds the Continuous Plankton Recorder Survey. M.E. was supported by NERC grants UK Changing Arctic programme DIAPOD and UK NERC CLASS. We would like to thank the owners and crews that have towed the CPRs on a voluntary basis for over 80 years contributing to one of the world's largest and longest ongoing ecological experiments. Without these early pioneers of citizen science and broad-scale volunteer monitoring projects this unique ecological dataset would never have been financially or logistically viable.

## Author contributions
M.E. conceived the project. P.H., E.G. and M.B. performed the numerical analysis of the data. M.E. wrote the manuscript. A.L., A.A. and G.T. provided expertise on euphausiids. All authors contributed to the discussion of the results and reviewed the manuscript.

## Competing interests
The authors declare no competing interests.
