## [Peer Review File · Communications Biology]

REVIEWERS' COMMENTS:

Reviewer #1 (Remarks to the Author):

This paper addresses the question of whether krill populations in the North Atlantic Ocean have experienced regionwide declines over the past 60 years due to poleward isotherm movement and habitat compression. The article uses the 60-year Continuous Plankton Recorder (CPR) dataset and associated physical data to examine that question. The topic of this paper is certainly of interest to a broad range of researchers studying range shifts, plankton communities, and marine food webs. The finding of regionwide krill declines, rather than poleward range shifts of the entire population, presents a novel angle on how marine species will be affected by climate change.

The manuscript in its current form represents substantial reorganization, clarity, and focus from its previous state. The main theme of long-term declines in regionwide krill populations across the North Atlantic, which are supported by the CPR data, is clear throughout. The authors have made substantial effort to address possible limitations of CPR sampling, but also present a convincing argument for why the trends are still legitimate. Introductions of supporting observations (e.g., changes in phytoplankton populations, other dominant zooplankton taxa) and potential explanatory climate indices (NAO, etc.) are well-timed and contextualized within the overall flow of the paper. The statistical analyses are overall well-described and sound.

My main critiques are as follows:

- 1) The authors should clarify throughout the study the sources they draw upon for euphausiid species habitat tolerances/preferences and changes in other zooplankton species. These sources are mentioned in the Methods but also need to be clarified briefly in the main text. It is currently not clear when other referenced zooplankton datasets come from CPR or external studies. Specific areas include:
 - Line 89 – Clarify where the 'taxonomic records' are from (i.e., 'see Methods for CPR taxonomic analyses')
 - Lines 91-97 – This section needs more references to substantiate its claims about *T. longicaudata* ('primarily governed by the thermal regime' – citation?), and it needs clarification about which statements and conclusions about *T. longicaudata* come from this study versus past experimental studies (i.e., on thermal tolerances)
 - Lines 169-177, etc. – another area that would benefit from clarification (e.g., these zooplankton measurements come from CPR)

- 2) Fig. 2 (decadal maps) would benefit from a box on each map that indicates the 248 core cells used to calculate krill range shifts in Fig. 3. As is, Fig. 2 is slightly misleading because it shows additional northern sampling in the final decade. Although the authors address in the Methods that they used the uniformly sampled region for their calculations, a box on each map and a statement in the Fig. 3 caption explaining from which region in Fig. 2 those numbers came would make Fig. 3 more clear and convincing (i.e., the populations didn't just shift very far north into an unsampled region).

- 3) I still feel that 'polar fronts limiting migrations' (Line 213-forward) needs more explanation and references, since it is the main justification for why cold, northern isotherms (and krill) aren't shifting poleward. I understand that this paper is focused on krill rather than atmospheric phenomena, but right now, 'polar fronts' sounds too speculative without clear references.

Line comments:

- Lines 56-57 – consider listing the 'three most common euphausiid species'
- Fig. 1a – I again suggest adding map labels for the commonly referenced sub-regions of the North Atlantic ('subpolar gyre, Norwegian Sea, etc.) for readers who are not at all familiar with this region.

- Line 173 – specify 'pelagic hyperiid amphipods' if this is the first time you mention them
- Lines 290-291 – specify 'meters' for depths, and move '(0-100)' after 'surface layers'
- Line 311 – specify units for lat & lon (60 x 20 what?)

Reviewer #2 (Remarks to the Author):

Mr Edwards and his co-authors present a comprehensive analysis of euphausiid data from the CPR survey from 1958 to 2017, relating changes in krill abundance to changes in sea surface temperature (SST) in the North Atlantic. They report a clear decline (by about 50%) in "northern krill" abundance over this 6-decade period, associated with regime shifts in the 1980s (North Sea) and the late 1990s (northwest Atlantic). Their SST analysis reveals that, while the southern boundary of the thermal habitat has moved northwards, the northern boundary remained constant due to strong oceanographic features. Edwards et al. conclude that the observed decline in krill abundance was associated with a reduction of the latitudinal extent of the thermal habitat of krill in the North Atlantic.

This study constitutes an important contribution to scientific discussions on the impact of the climate crisis on oceanic ecosystems, highlighting that poleward shifts following ocean warming are not always linear (my personal opinion is that they hardly ever are). By laying a focus on euphausiids, which are ecologically highly relevant but in the northern hemisphere do not receive the same scientific attention as in the south, this study presents a new perspective on ecological change in the north Atlantic, with major implications for predictions on fish distribution and marine management. The time period covered of this study is impressive, and demonstrates the high value of repeated measurements ("monitoring") in marine ecosystems.

The results and supporting arguments are presented in a very clear language, and the figures show very convincingly the patterns observed by this study.

I am in full support of this study and recommend swift publication. However, there is no perfection in science. I acknowledge that this is a resubmission, and many aspects of potential improvement have already been addressed by the authors. However, the fact that I still see the same issues as previous reviewers highlights that some (minor) improvement is still recommended:

Taxonomy. This revised manuscript is still somewhat vague in its taxonomic scope. In the introduction (LL34/35), "northern krill" and "Antarctic krill" are introduced, without a taxonomic definition of the species included in these terms. Please make this visible by adding the scientific names (e.g. *Euphausia superba*) in brackets. Throughout the paper, it is not clear which species the authors mean. In the methods section, they explain that a) euphausiids are not identified to species level in CPR analysis, and b) *Thysanoessa longicauda* might be the dominant species at hand.

This low taxonomic resolution is a bit unfortunate, because the temperature envelope of euphausiids is the main topic of this paper, and temperature optima and tolerances differ between species. Besides a shift in abundance, one may also expect a shift in species composition over this long time scale. It would be good if the authors addressed these issues in a revised version by stating the taxonomic scope (and limitations) more clearly in the main text, and by presenting analyses based on subsamples from different phases of the observation period, which can be considered representative of the taxonomic composition and its potential changes over time.

Depth distribution. The CPR samples at 10 meter water depth, and thus does not inform about the distribution of animals in deeper water layers. I do agree with the authors that at large sample sizes (as is the case here) variability in CPR data can be considered to mirror relative changes in abundance over the whole depth distribution of a species. The example of *Calanus finmarchicus* presented to support this assumption, however, is nothing more than just an example. It is not a causal argument to assume the same for the different krill species. My point here is that besides latitudinal shifts, euphausiids may react to SST increase by vertical shifts in their distribution. The CPR cannot monitor this, but I would appreciate a more "in-depth" discussion of this aspect in the main text, based on available literature. The argument made by the authors that a deeper distribution (below the mixed layer) would cut them off from phytoplankton food may or may not be true. The euphotic zone extends down to 200 m, and phytoplankton may also be abundant at greater depth. Besides, heterotrophic food may be an abundant carbon source independent of light, especially for the omnivorous *Thysanoessa* spp.. Furthermore, if the CPR sampling is biased

towards juvenile krill as stated by the authors, both latitudinal and vertical distributions may differ in adults. For example, we found *Thysanoessa longicaudata* and *T. inermis* in $< 0^{\circ}$ surface water under the sea ice of the Arctic Ocean. These considerations do not diminish the value of the presented data. But I believe they should be addressed clearly in the main text.

Management implications. In my view, the results of this study have major implications for ecosystem studies and marine management. Just as an example: If euphausiids declined by 50%, must we also expect a decline in pelagic fishes feeding on them? May "borealization" observed in Arctic fish communities not only be driven by temperature rise in Arctic waters, but also by food scarcity in boreal waters? What does this imply for the management advice by ICES?

Statistics. I would appreciate a more comprehensive explanation of the PCA method. To those readers not performing multivariate statistics in their daily work, the concept of eigenvalue and principal components are unfamiliar. I do work with these methods, but even for me it is still not clear which kind of matrix the authors used in their analysis (i.e. which were the rows and columns in their raw data).

Data availability. The formulation "The data that support the findings in this study are available from the corresponding author upon reasonable request" is not in line with the commitment of the scientific community to open data sharing, as we practice it today. Besides the fact that "reasonable request" is not defined, nowadays data are usually made available in online repositories, such as Pagaea or OBIS. I am sure it is not difficult to refer to institutional or national/international repositories, as the data of SST and CPR are already available somehow.

Spelling. I found a couple of typos and potential comma errors reading the manuscript and recommend thorough proof-reading before resubmission.

Reviewer #3 (Remarks to the Author):

Dear Authors,

your manuscript compiles an impressive data set on euphausiid distribution in the North Atlantic over 60 years as determined using the CPR. You outline the limitations of CPR studies (being confined to the upper 100m of the water column) in your manuscript, and I agree that despite the bias which comes with the method, your approach indeed allows to tackle the question you ask, i.e. how does the distribution of euphausiids change in relation to SST.

The figures 1 and 2 very nicely show that the centre of the distribution is the North West Atlantic, and in that region the abundance decreases tremendously according to your data. As krill is an important component of the pelagic food webs, such information is of interest to the scientific community, and potentially also for fisheries.

The main conclusion is that the decrease in krill abundance in the North Atlantic can be explained by the northward shift of the 12/13 C isobare whereas the 7/8 C isobare which presents the lower temperature limit of the dominant euphausiid species *Thysanoessa longicornis* is relatively stable, leading to a reduction of the area, suitable for krill.

Despite the limitations of the sampling, restricted to 10m depth, the data are very interesting and clearly demonstrate that the abundance of krill has decreased in that layer. What is missing, however, is the discussion on what is to be expected considering deeper waters, i.e. temperature increase is most distinct in the upper meters but less pronounced in deeper layers. Krill could thus migrate little deeper, avoid the higher temperatures at the surface with possible consequences for food availability. I would like to see this aspect discussed more thoroughly.

To me, the characterization of the North Atlantic and its ecosystems is not fully clear. Frequently, the authors refer to the North Atlantic basin, while, in fact, there are several basins. I was also surprised to read (line 198 ff), that the sub-polar regions of the North Atlantic are subject to Atlantification – this reads a bit odd and may need a little of explanation, i.e. region where Polar and Atlantic water masses mix, saline and warmer water masses increase...The next sentences address the Arctic Seas, and I do not fully understand the cause and effects (sea-ice loss? Not so much in the North Atlantic, right?), you address here. I also do not fully understand why you address the Arctic

and the changes in the species inventory here, thus the relevance to your study/data set.

You argue that euphausiids are "squeezed" into a smaller area, however, if that would be the case, shouldn't we observe abundances as high as in 1958-1967 in that area between the min and max temperature? To me it appears that the abundance has decreased everywhere, also in that area where the temperature theoretically "suits" the euphausiids. E.g. in the 2008-2017, there is only one spot, south-east off Greenland, where the abundances seem as high as they have been fifty years before. Thus, temperature should not be the only reason why euphausiids decrease in abundance, correct? I think this should be discussed. I would also appreciate more detailed information on the statistical analyses you have performed.

Detailed comments:

Line 34: Total mass may be similar, however, the Southern Ocean is much more defined and smaller than the North Atlantic Ocean.

Line 35: I wonder whether there more appropriate references than #1, 5 and 7? These are rather general papers on krill biology and distribution and do not specifically address fisheries.

Line 53: „vast majority of krill abundance“ I am not a native speaker but to me it would be „vast majority of krill“ only or „highest abundances of krill“

Line 69 „specific measure were taken“ – should be measureS were?

Line 140 „there have been ...evolution“ – should be there HAS been?

Lines 172-178: which data (copepod species , amphipods and Appendicularia) do you refer to? There is no reference. Own unpublished data?

Line 188: "this group appear" – should be this group appears

Line 208: "ocean temperature rise" – should be ocean temperatures rise or ocean T rises

Authors response to reviewers comments

REVIEWERS' COMMENTS:

Reviewer #1 (Remarks to the Author):

This paper addresses the question of whether krill populations in the North Atlantic Ocean have experienced regionwide declines over the past 60 years due to poleward isotherm movement and habitat compression. The article uses the 60-year Continuous Plankton Recorder (CPR) dataset and associated physical data to examine that question. The topic of this paper is certainly of interest to a broad range of researchers studying range shifts, plankton communities, and marine food webs. The finding of regionwide krill declines, rather than poleward range shifts of the entire population, presents a novel angle on how marine species will be affected by climate change.

The manuscript in its current form represents substantial reorganization, clarity, and focus from its previous state. The main theme of long-term declines in regionwide krill populations across the North Atlantic, which are supported by the CPR data, is clear throughout. The authors have made substantial effort to address possible limitations of CPR sampling, but also present a convincing argument for why the trends are still legitimate. Introductions of supporting observations (e.g., changes in phytoplankton populations, other dominant zooplankton taxa) and potential explanatory climate indices (NAO, etc.) are well-timed and contextualized within the overall flow of the paper. The statistical analyses are overall well-described and sound.

My main critiques are as follows:

1) The authors should clarify throughout the study the sources they draw upon for euphausiid species habitat tolerances/preferences and changes in other zooplankton species. These sources are mentioned in the Methods but also need to be clarified briefly in the main text. It is currently not clear when other referenced zooplankton datasets come from CPR or external studies. Specific areas include:

- Line 89 – Clarify where the ‘taxonomic records’ are from (i.e., ‘see Methods for CPR taxonomic analyses’)

Added from CPR data in the text

- Lines 91-97 – This section needs more references to substantiate its claims about *T. longicaudata* (‘primarily governed by the thermal regime’ – citation?), and it needs clarification about which statements and conclusions about *T. longicaudata* come from this study versus past experimental studies (i.e., on thermal tolerances)

This statement is mainly based on the work by and which is now referenced: Williams, R. & Lindley, J. A. Variability in abundance, vertical distribution and ontogenetic migrations of *Thysanoessa longicaudata* (Crustacea: Euphausiacea) in the north-eastern Atlantic Ocean. *Mar. Biol.* **69**, 321–330 (1982).

- Lines 169-177, etc. – another area that would benefit from clarification (e.g., these zooplankton measurements come from CPR)

Added: Data from the CPR Survey show....

2) Fig. 2 (decadal maps) would benefit from a box on each map that indicates the 248 core cells used to calculate krill range shifts in Fig. 3. As is, Fig. 2 is slightly misleading because it shows additional northern sampling in the final decade. Although the authors address in the Methods that they used the uniformly sampled region for their calculations, a box on each map and a statement in the Fig. 3 caption explaining from which region in Fig. 2 those numbers came would make Fig. 3 more clear and convincing (i.e., the populations didn't just shift very far north into an unsampled region).

While this is a very good idea this was too difficult to implement and keep all the information legible so we have kept the original figure unchanged.

3) I still feel that 'polar fronts limiting migrations' (Line 213-forward) needs more explanation and references, since it is the main justification for why cold, northern isotherms (and krill) aren't shifting poleward. I understand that this paper is focused on krill rather than atmospheric phenomena, but right now, 'polar fronts' sounds too speculative without clear references.

We have now added the following reference to give context

From the spatial maps of krill it can be seen that the main distribution of abundance is found in the North Atlantic subpolar gyre. Despite annual and decadal variations in the shape of the North Atlantic subpolar gyre it remains quite spatially robust with the northerly boundary remaining fairly static over a multidecadal period (see figure below from Biri & Klein, 2019). This is unlike the North East Atlantic where isotherms have been moving northward over the last 50 years.

Biri, S. & Klein, B. 2019. North Atlantic sub-polar gyre climate index: a new approach. *Journal of Geophysical research: oceans* 124: 4222-4237.

Annual variations in the North Atlantic subpolar gyre system (from Biri & Klein, 2019)

Line comments:

- Lines 56-57 – consider listing the ‘three most common euphausiid species’

We have referred to the Methods for the species information

- Fig. 1a – I again suggest adding map labels for the commonly referenced sub-regions of the North Atlantic (‘subpolar gyre, Norwegian Sea, etc.) for readers who are not at all familiar with this region.

We considered this would make the figure difficult to read due to the size of the figure and the already added grids

- Line 173 – specify ‘pelagic hyperiid amphipods’ if this is the first time you mention them

Added (amphipoda) to text

- Lines 290-291 – specify ‘meters’ for depths, and move ‘(0-100)’ after ‘surface layers’

Added

- Line 311 – specify units for lat & lon (60 x 20 what?)

Added: total geographic size....

Reviewer #2 (Remarks to the Author):

Mr Edwards and his co-authors present a comprehensive analysis of euphausiid data from the CPR survey from 1958 to 2017, relating changes in krill abundance to changes in sea surface temperature (SST) in the North Atlantic. They report a clear decline (by about 50%) in “northern krill” abundance over this 6-decade period, associated with regime shifts in the 1980s (North Sea) and the late 1990s (northwest Atlantic). Their SST analysis reveals that, while the southern boundary of the thermal habitat has moved northwards, the northern boundary remained constant due to strong oceanographic features. Edwards et al. conclude that the observed decline in krill abundance was associated with a reduction of the latitudinal extent of the thermal habitat of krill in the North Atlantic.

This study constitutes an important contribution to scientific discussions on the impact of the climate crisis on oceanic ecosystems, highlighting that poleward shifts following ocean warming are not always linear (my personal opinion is that they hardly ever are). By laying a focus on euphausiids, which are ecologically highly relevant but in the northern hemisphere do not receive the same scientific attention as in the south, this study presents a new perspective on ecological change in the north Atlantic, with major implications for predictions on fish distribution and marine management. The time period covered of this study is impressive, and demonstrates the high value of repeated measurements (“monitoring”) in marine ecosystems.

The results and supporting arguments are presented in a very clear language, and the figures show very convincingly the patterns observed by this study.

I am in full support of this study and recommend swift publication. However, there is no perfection in science. I acknowledge that this is a resubmission, and many aspects of potential improvement have already been addressed by the authors. However, the fact that I still see

the same issues as previous reviewers highlights that some (minor) improvement is still recommended:

Taxonomy. This revised manuscript is still somewhat vague in its taxonomic scope. In the introduction (LL34/35), “northern krill” and “Antarctic krill” are introduced, without a taxonomic definition of the species included in these terms. Please make this visible by adding the scientific names (e.g. *Euphausia superba*) in brackets. Throughout the paper, it is not clear which species the authors mean. In the methods section, they explain that a) euphausiids are not identified to species level in CPR analysis, and b) *Thysanoessa longicaudata* might be the dominant species at hand.

We have now added the scientific names

This low taxonomic resolution is a bit unfortunate, because the temperature envelope of euphausiids is the main topic of this paper, and temperature optima and tolerances differ between species. Besides a shift in abundance, one may also expect a shift in species composition over this long time scale. It would be good if the authors addressed these issues in a revised version by stating the taxonomic scope (and limitations) more clearly in the main text, and by presenting analyses based on subsamples from different phases of the observation period, which can be considered representative of the taxonomic composition and its potential changes over time.

We have now addressed this in the section on ‘potential biases and limitations’.

Depth distribution. The CPR samples at 10 meter water depth, and thus does not inform about the distribution of animals in deeper water layers. I do agree with the authors that at large sample sizes (as is the case here) variability in CPR data can be considered to mirror relative changes in abundance over the whole depth distribution of a species. The example of *Calanus finmarchicus* presented to support this assumption, however, is nothing more than just an example. It is not a causal argument to assume the same for the different krill species. My point here is that besides latitudinal shifts, euphausiids may react to SST increase by vertical shifts in their distribution. The CPR cannot monitor this, but I would appreciate a more “in-depth” discussion of this aspect in the main text, based on available literature. The argument made by the authors that a deeper distribution (below the mixed layer) would cut them off from phytoplankton food may or may not be true. The euphotic zone extends down to 200 m, and phytoplankton may also be abundant at greater depth. Besides, heterotrophic food may be an abundant carbon source independent of light, especially for the omnivorous *Thysanoessa* spp.. Furthermore, if the CPR sampling is biased towards juvenile krill as stated by the authors, both latitudinal and vertical distributions may differ in adults. For example, we found *Thysanoessa longicaudata* and *T. inermis* in $< 0^\circ$ surface water under the sea ice of the Arctic Ocean. These considerations do not diminish the value of the presented data. But I believe they should be addressed clearly in the main text.

Management implications. In my view, the results of this study have major implications for ecosystem studies and marine management. Just as an example: If euphausiids declined by 50%, must we also expect a decline in pelagic fishes feeding on them? May “borealization” observed in Arctic fish communities not only be driven by temperature rise in Arctic waters, but also by food scarcity in boreal waters? What does this imply for the management advice by ICES?

Although we cannot absolutely refute the hypothesis of the reviewer that euphausiids in recent years may have assumed a deeper distribution, below the sampling depth of the CPR, we consider this unlikely mainly due to that the upper mixed layer, across much of the North Atlantic, spans the upper 30 to 40 m over which the temperature is relatively constant. To remain within a different (colder) thermal regime, euphausiids must completely avoid this layer, assuming a distribution that is always deeper than 40 m, making the productivity that is concentrated in the upper mixed layer (and within the euphotic zone) inaccessible to them. Residing in the non-productive layers will make it extremely challenging for euphausiids to meet their energy budget. Hence, euphausiids avoiding the upper mixed layer and adopting a deeper distribution will be at a severe disadvantage and are likely to become locally extinct.

We have addressed the limitations in CPR sampling as it is biased towards juvenile krill as in the section ‘potential limitations and biases’.

We agree that the 50% decline in euphausiid populations across the North Atlantic will have major implications for the rest of the food web including fish stocks, however, we have not speculated further than this as this is beyond the scope of the current study. We definitely agree that this study would be very important to fisheries managers and for scientists involved in the management advice for ICES.

Statistics. I would appreciate a more comprehensive explanation of the PCA method. To those readers not performing multivariate statistics in their daily work, the concept of eigenvalue and principal components are unfamiliar. I do work with these methods, but even for me it is still not clear which kind of matrix the authors used in their analysis (i.e. which were the rows and columns in their raw data).

We refer the referee to the papers below which goes into much more technical detail on PCA methods for analysing this kind of multivariate data. For the question the referee asks referring to the rows and columns in the raw data they initially are a columns and rows consisting of the longitude and latitude of the variable (krill abundance) and time (day/month/year). This raw data is then converted to a gridded format (using spatial interpolation) into a matrix that consists of a 3-way matrix 60 years (period 1958 to 2017) × (20 longitudes x 60 latitudes).

Beaugrand, G., Edwards, M., Brander, K., Luczak, C. & Ibanez, F. Causes and projections of abrupt climate-driven ecosystem shifts in the North Atlantic. *Ecol. Lett.* **11**, 1157–68 (2008).

Harris, V., Edwards, M. & Olhede, S. C. Multidecadal Atlantic climate variability and its impact on marine pelagic communities. *J. Mar. Syst.* **133**, 55–69 (2014).

Harris, V., Olhede, S. C. & Edwards, M. Multidecadal spatial reorganisation of plankton communities in the North East Atlantic. *J. Mar. Syst.* **142**, 16–24 (2015).

Data availability.

The formulation “The data that support the findings in this study are available from the corresponding author upon reasonable request” is not in line with the commitment of the scientific community to open data sharing, as we practice it today. Besides the fact that “reasonable request” is not defined, nowadays data are usually made available in online repositories, such as Pagaia or OBIS. I am sure it is not difficult to refer to institutional or

national/international repositories, as the data of SST and CPR are already available somehow. We have removed this sentence. All raw data is available by request and the data is now open source and available at the open data portal see: <https://www.mba.ac.uk/blog/oceans-open-data>

Spelling. I found a couple of typos and potential comma errors reading the manuscript and recommend thorough proof-reading before resubmission.
checked

Reviewer #3 (Remarks to the Author):

Dear Authors,

your manuscript compiles an impressive data set on euphausiid distribution in the North Atlantic over 60 years as determined using the CPR. You outline the limitations of CPR studies (being confined to the upper 100m of the water column) in your manuscript, and I agree that despite the bias which comes with the method, your approach indeed allows to tackle the question you ask, i.e. how does the distribution of euphausiids change in relation to SST.

The figures 1 and 2 very nicely show that the centre of the distribution is the North West Atlantic, and in that region the abundance decreases tremendously according to your data. As krill is an important component of the pelagic food webs, such information is of interest to the scientific community, and potentially also for fisheries.

The main conclusion is that the decrease in krill abundance in the North Atlantic can be explained by the northward shift of the 12/13 °C isobare whereas the 7/8 °C isobare which presents the lower temperature limit of the dominant euphausiid species *Thysanoessa longicornis* is relatively stable, leading to a reduction of the area, suitable for krill.

Despite the limitations of the sampling, restricted to 10m depth, the data are very interesting and clearly demonstrate that the abundance of krill has decreased in that layer. What is missing, however, is the discussion on what is to be expected considering deeper waters, i.e. temperature increase is most distinct in the upper meters but less pronounced in deeper layers. Krill could thus migrate little deeper, avoid the higher temperatures at the surface with possible consequences for food availability. I would like to see this aspect discussed more thoroughly.

Although we cannot absolutely refute the hypothesis of the reviewer that euphausiids in recent years may have assumed a deeper distribution, below the sampling depth of the CPR, we consider this unlikely for two reasons:-

1. The upper mixed layer, across much of the North Atlantic, spans the upper 30 to 40 m over which the temperature is relatively constant. To remain within a different (colder) thermal regime, euphausiids must completely avoid this layer, assuming a distribution that is always deeper than 40 m, making the productivity that is concentrated in the upper mixed layer (and within the euphotic zone) inaccessible to them. Residing in the non-productive layers will make it extremely challenging for euphausiids to meet their energy budget. Hence, euphausiids avoiding the upper mixed layer and adopting a deeper distribution will be at a severe disadvantage and are likely to become locally extinct

2. Although the thermal regime may have changed over recent decades, the day/night cycle, which acts as both a cue and a driver for their various diel vertical migration patterns, has not changed and we expect this behaviour to remain the same. This means that any sampling bias caused by DVM will be the same across decades irrespective of any thermal shift.

To me, the characterization of the North Atlantic and its ecosystems is not fully clear. Frequently, the authors refer to the North Atlantic basin, while, in fact, there are several basins. I was also surprised to read (line 198 ff), that the sub-polar regions of the North Atlantic are subject to Atlantification – this reads a bit odd and may need a little of explanation, i.e. region where Polar and Atlantic water masses mix, saline and warmer water masses increase... The next sentences address the Arctic Seas, and I do not fully understand the cause and effects (sea-ice loss? Not so much in the North Atlantic, right?), you address here. I also do not fully understand why you address the Arctic and the changes in the species inventory here, thus the relevance to your study/data set.

We have looked at and cleared some of the text in reference to the North Atlantic names. The recent phenomenon of the ‘Atlantification’ of more northerly Atlantic regions refers to more warmer and saline waters becoming more prevalent in traditionally cooler regions (e.g. the Barents Sea). The main reference to this is:
Polyakov, I. V. *et al.* Greater role for Atlantic inflows on sea-ice loss in the Eurasian Basin of the Arctic Ocean. *Science* (80-.). **356**, 285–291 (2017).

You argue that euphausiids are “squeezed” into a smaller area, however, if that would be the case, shouldn’t we observe abundances as high as in 1958-1967 in that area between the min and max temperature? To me it appears that the abundance has decreased everywhere, also in that area where the temperature theoretically “suits” the euphausiids. E.g. in the 2008-2017, there is only one spot, south-east off Greenland, where the abundances seem as high as they have been fifty years before. Thus, temperature should not be the only reason why euphausiids decrease in abundance, correct? I think this should be discussed. I would also appreciate more detailed information on the statistical analyses you have performed.

We agree that temperature alone does not explain all the patterns observed and that other trophic interactions would also be important. For this study, however, trophic interactions were not considered as the focus was on the relationship between long-term patterns and climate change in the form of multi-decadal indices. We also realise that the mechanisms for all these patterns observed cannot be fully explained by temperature and would warrant further investigation. Although the following paper (Barange, et al 2009) discusses fish stocks it does have some interesting information on how habitat expansions and contractions affect different populations. The study implies that as small pelagic fish populations grow in stock size both the area they occupy and their packing density increase and become more spatially fragmented (of course in this paper’s case we would have to take it the other way around). In this study we are looking at a converse situation happening over time for the Western North Atlantic euphausiids, and if this is partially explained by a basin model (as well as thermal habitat preferences), it would mean that as their total population size shrinks or is squeezed, so does the geographical range of the population, and so does the mean density, even in their thermal optimum.

Barange, M. et al (2009). Habitat expansion and contraction in anchovy and sardine populations. *Progress in Oceanography*, 83: 251-260.

In terms of the statistical analysis for the interpolation of the maps. In this study we have used spatial interpolation methods to reduce sampling biases in space and time over the large area of the North Atlantic. Biological data were regularised on a 2° by 2° grid at a monthly scale for the period 1958 - 2017 by using an Inverse Distance Weighting (IDW) interpolation. A maximum search radius of 250 km was set as well as a minimum number of neighbours of 5. This has allowed us to reduce the biases associated with the changes in sampling over time. The spatial method we used also allows us to estimate with a degree of confidence the interpolated data points. If a certain degree of confidence was not met, we have discarded the spatially interpolated data. However, there are also circumstances where you still get statistical artefacts depending on sampling with extreme high and low values that can lead to spatial anomalies this could be the case for the southern Greenland area in the 2008-2017 period.

This had been the recommended interpolation method for CPR data at the North Atlantic scale where a 2 by 2 degree grid is used. This method has been used in numerous publications and has most recently used in Beaugrand et al 2019 where we have looked at the scale of the North Atlantic and used a 2 by 2 grid for interpolation.

Beaugrand, G., Edwards, M & Helaouet, P. 2019. An ecological partition of the Atlantic Ocean and its adjacent seas. *Progress in Oceanography*, 173: 86-102.

Detailed comments:

Line 34: Total mass may be similar, however, the Southern Ocean is much more defined and smaller than the North Atlantic Ocean.

OK

Line 35: I wonder whether there more appropriate references than #1, 5 and 7? These are rather general papers on krill biology and distribution and do not specifically address fisheries.

Yes the referee is correct as they are more general references but they do reference fisheries within the text

Line 53: „vast majority of krill abundance“ I am not a native speaker but to me it would be „vast majority of krill“ only or „highest abundances of krill“

Changed the sentence to make it read more clearly.

Line 69 „specific measure were taken“ – should be measureS were?

Added ‘measures’

Line 140 „there have been ... evolution“ – should be there HAS been?

Changed to ‘has’

Lines 172-178: which data (copepod species , amphipods and Appendicularia) do you refer to? There is no reference. Own unpublished data?

Now referred to as CPR data

Line 188: “this group appear” – should be this group appears
changed

Line 208: “ocean temperature rise” – should be ocean temperatures rise or ocean T rises
changed